# DEBUNK THE MYTH OF SFT GENERALIZATION

## ABSTRACT

A prevailing view holds that supervised fine-tuning (SFT) memorizes training data and fails to generalize, whereas reinforcement learning (RL) attains broader robustness. We revisit this claim through a systematic evaluation on two decision-making benchmarks, *Sokoban* and *General Points*, and arrive at a different conclusion. We show that much of SFT's perceived failure stems from *frozen-prompt* artifacts: when trained on fixed instruction templates, SFT models cling to training semantics rather than adapting to new ones. Introducing *prompt diversity* during training breaks this shortcut and yields strong generalization to unseen instruction variants without harming in-distribution performance. Beyond instruction shifts, we ask whether SFT can generalize to strictly harder tasks. Here, *chain-of-thought (CoT) supervision* provides an algorithmic scaffold that markedly improves transfer to more difficult regimes, such as larger *Sokoban* grids with additional boxes and arithmetic with out-of-distribution values or five-card compositions that increase combinatorial complexity. Finally, combining prompt diversity with CoT achieves the best of both worlds: robust generalization across both instruction-variant and difficulty-variant settings, matching or surpassing RL baselines on our benchmarks while retaining SFT's simplicity and stability. These findings challenge the narrative that SFT is inherently inferior to RL and support a data-centric perspective: with appropriately curated demonstrations, vanilla SFT can generalize as strongly as RL.

## 1 INTRODUCTION AND BACKGROUND

Supervised fine-tuning (SFT) has emerged as one of the most widely adopted post-training approaches for large language models (LLMs)(Ouyang et al., 2022; Hu et al., 2022; Wang et al., 2022; Touvron et al., 2023). Its popularity is driven by several practical advantages: the maximum-likelihood objective is straightforward to optimize, it can leverage abundant off-policy demonstrations, and it is substantially more cost-effective and stable than reinforcement learning (RL)-based finetuning, the primary alternative. For these reasons, SFT remains the backbone of many widely deployed instruction-tuned and alignment-focused models.

At the same time, the very properties that make SFT appealing have also been associated with perceived limitations. Standard cross-entropy training optimizes the likelihood of reference outputs rather than downstream task reward. This objective has been argued to encourage overfitting to surface forms, collapse output diversity, and induce unnecessary drift away from the pretrained base model when training data are narrow (Li et al., 2024; Xiao, 2024). Consequently, a series of studies have reported limited generalization performance of SFT compared to RL-based finetuning (Hu et al., 2025; Li et al., 2024; Xiao, 2024; Shenfeld et al., 2025; Jin et al., 2025). Following this narrative, considerable effort has been invested in designing algorithmic modifications—ranging from data reweighting to proximal objectives—to improve the generalization capacity of SFT (Li et al., 2024; Qin & Springenberg, 2025; Wu et al., 2025; Zhu et al., 2025).

This line of work raises an important question: are the limitations of SFT intrinsic to the maximum-likelihood objective, or do they primarily reflect shortcomings in data design and evaluation? Addressing this question is critical for understanding the trade-offs between SFT and RL-based methods. RL offers the appeal of directly optimizing task-aligned rewards but is often substantially more complex, unstable, and resource-intensive to apply at scale. If vanilla SFT can, under the right conditions, achieve strong generalization, then its simplicity and efficiency may offer a more practical path forward for many applications.

In this work, we revisit the generalization capacity of SFT and arrive at a different conclusion from much of the prior literature. **We show that vanilla SFT, without any algorithmic modification, can generalize substantially better than previously reported.** Our findings hold across both single-turn and multi-turn decision-making tasks. A central insight of this study is that data quality plays a more decisive role than algorithmic adjustments: in particular, the combination of prompt diversity and chain-of-thought (CoT) supervision is sufficient to enable SFT to generalize robustly, not only to tasks with close resemblance to training data but also to tasks of significantly greater difficulty.

In summary, this work makes the following contributions:

- **Re-evaluation of SFT generalization:** We provide a systematic study of vanilla SFT's generalization capacity across multiple domains and base models, demonstrating performance that challenges prevailing perceptions.

- **Identification of data-centric drivers:** We show that prompt diversity and CoT supervision serve as a sufficient recipe for strong generalization, often outweighing the need for algorithmic modifications

- **Reframing of the SFT–RL trade-off.** We present evidence that, given appropriate data design, vanilla SFT is competitive with RL-based finetuning on our benchmarks, and we discuss implications for method selection when optimizing for simplicity, stability, and cost.

While our findings highlight the potential of vanilla SFT, we note that our evaluation is limited to decision-making tasks, and further investigation is required to assess whether similar trends hold in more open-ended or creative generation settings. The remainder of the paper is organized as follows. Section 2 reviews prior work on evaluating, analyzing, and modifying SFT. Section 3 provides the necessary technical background. Section 4 describes the experimental setup. Sections 5 and 6 present our main empirical findings. Section 7 concludes with a summary and discussion of potential future directions.

## 2 RELATED WORK

### 2.1 SUPERVISED FINE-TUNING VS. REINFORCEMENT LEARNING

Fine-tuning methods for LLMs largely fall into two categories: SFT and RL. SFT aligns a base model by minimizing the negative log-likelihood on off-policy demonstrations. This objective is straightforward to optimize and typically requires fewer computational resources than RL. However, the vanilla cross-entropy loss is often perceived to encourage memorization rather than the acquisition of generalizable capabilities (Li et al., 2024; Xiao, 2024).

By contrast, RL fine-tuning relies on the base model's ability to explore and generate rewarding trajectories without explicit step-by-step supervision. Progress therefore depends on producing sufficiently good rollouts, and RL that is not warm-started from strong demonstrations often stalls on harder domains where SFT excels (Hu et al., 2025). Given these complementary strengths and weaknesses, a growing body of work has directly compared SFT and RL under controlled settings. Prior studies report that SFT tends to memorize training prompts and degrade on out-of-distribution (OOD) performance across both text and vision, whereas RL improves both in-distribution (ID) and OOD performance (Chu et al., 2025).

Huan et al. (2025) show that RL-tuned models trained on mathematical tasks transfer positively to other reasoning and non-reasoning domains, while SFT-tuned models often incur negative transfer on non-reasoning domains. Their PCA and KL-divergence analyses indicate that RL-tuned policies remain closer to the base model than SFT-tuned ones. Similarly, Shenfeld et al. (2025) find that SFT suffers greater forgetting than RL, measurable as larger distributional shift when adapting to new tasks; due to its on-policy nature, RL tends to stay nearer to its base policy. Finally, Jin et al. (2025) argue that SFT's limited generalization arises from rigid alignment of parameter directions to the target task, which over-specializes the model, while RL can reorient parameter subspaces toward more robust configurations.

While these studies conclude that RL generalizes better than SFT, we revisit this narrative and show that vanilla SFT can achieve comparable or even superior generalization when trained with appropri-

ate data. Our findings suggest that many reported limitations of SFT stem not from the maximum-likelihood objective itself but from restricted data design.

## 2.2 REFINEMENT ON SUPERVISED FINE-TUNING

Supervised fine-tuning remains the most prominent post-training method for LLMs because it aligns a base model with a simple maximum-likelihood objective, leverages abundant off-policy demonstrations, and is substantially cheaper and easier to optimize than RL. Yet these same properties also expose its limitations: vanilla cross-entropy optimizes the likelihood of reference demonstrations rather than task reward, may overfit surface forms and collapse output diversity, and can induce unnecessary drift away from the pretrained basin when data are narrow.

Motivated by these trade-offs, a growing literature refines SFT along axes of diversity preservation, objective reweighting, and proximal update control, with the goal of improving generalization and providing a stronger initialization for subsequent RL. For instance, Li et al. (2024) propose GEM, a game-theoretic SFT algorithm that preserves output diversity and benefits both test-time scaling and subsequent RL post-training. Qin & Springenberg (2025) interpret SFT as maximizing a loose lower bound on the RL objective whose tightness degrades as the policy drifts from a reference; they tighten this bound via importance reweighting that assigns higher weights to selected trajectories. Wu et al. (2025) reinterpret cross-entropy as a misspecified on-policy policy-gradient objective and propose a reweighting correction that improves generalization. Inspired by PPO, Zhu et al. (2025) incorporate a clipping mechanism into the SFT loss to discourage overconfident updates, preserve entropy, and improve generalization.

These refinements modify the SFT objective to address perceived weaknesses. By contrast, we show that unmodified SFT, when combined with prompt diversity and chain-of-thought demonstrations, already achieves strong generalization. Our results suggest that data design alone can close much of the gap previously attributed to deficiencies in the objective.

## 3 PRELIMINARIES

In this section, we briefly introduce the two algorithms that form the basis of our evaluation.

**Supervised Fine-Tuning (SFT).** Let $\mathcal{D} = \{(x, y^*)\}$ denote a corpus of expert demonstrations, where $x$ is a query and $y^*$ is the reference response. The objective of SFT is to minimize the negative log-likelihood (NLL) of the reference under the model distribution,

$$\mathcal{L}_{\text{SFT}}(\theta) = \mathbb{E}_{(x,y^*)\sim\mathcal{D}}\big[ -\log \pi_\theta(y^* \mid x)\big],$$

where $\pi_\theta$ denotes the policy (i.e., the LLM parameterized by $\theta$). This objective encourages the model to imitate reference demonstrations.

**Reinforcement Learning (RL) Fine-Tuning.** In contrast to SFT, RL fine-tunes the policy $\pi_\theta$ by directly optimizing for a scalar reward. Let $r(x, y)$ denote the reward assigned to a response $y$ given a query $x$. The objective is to maximize expected reward:

$$\mathcal{L}_{\text{RL}}(\theta) = \mathbb{E}_{x\sim\mathcal{D}, y\sim\pi_\theta(\cdot|x)}\big[r(x, y)\big].$$

In this work, we employ *Group Relative Policy Optimization* (GRPO) (Shao et al., 2024), a variant of Proximal Policy Optimization (PPO) (Schulman et al., 2017). GRPO retains the clipping mechanism of PPO but replaces the critic network with a group-based advantage estimator. Specifically, for each query $x$, a group of responses $\{y_i\}_{i=1}^G$ is sampled from the old policy $\pi_{\theta_{\text{old}}}(\cdot \mid x)$. Each response is scored with a reward $r_i$, and the normalized group-relative advantage is computed as

$$\hat{A}_i = \frac{r_i - \frac{1}{G}\sum_{j=1}^G r_j}{\text{std}(r_1, \ldots, r_G)}.$$

The GRPO objective is then

$$\mathcal{L}_{\text{GRPO}}(\theta) = \mathbb{E}_{x,\{y_i\}\sim\pi_{\theta_{\text{old}}}}\left[\frac{1}{G}\sum_{i=1}^G \min\left(\frac{\pi_\theta(y_i \mid x)}{\pi_{\theta_{\text{old}}}(y_i \mid x)}\hat{A}_i,\ \text{clip}_\epsilon\left(\frac{\pi_\theta(y_i \mid x)}{\pi_{\theta_{\text{old}}}(y_i \mid x)}, 1-\epsilon, 1+\epsilon\right)\hat{A}_i\right)\right].$$

# 4 EVALUATION TASKS AND DATA CONSTRUCTION

## 4.1 EVALUATION TASKS

We evaluate generalization on two tasks that expose both instruction variations and difficulty variations, a design partially inspired by Huang et al. (2025). *Sokoban* is a multi-step puzzle environment requiring long-horizon planning to avoid dead-ends, while *General Points* is an arithmetic reasoning task.

### 4.1.1 SOKOBAN

*Sokoban* is a deterministic puzzle-based environment where the objective is to push boxes onto designated targets within a step limit. The agent moves in four directions (Up, Right, Down, Left) on a grid world. Since boxes can only be pushed forward and not pulled back, actions are irreversible and require careful planning to avoid dead-ends.

**Instruction variations.** To test whether the model simply memorizes training prompts, we vary action instructions without changing task difficulty. The model is trained on canonical commands ("up, down, left, right") and evaluated on the following unseen mappings:

- *SimpleSokobanNumerical:* 1 = up, 2 = down, 3 = left, 4 = right.
- *SimpleSokobanAlphabetical:* A = up, B = down, C = left, D = right.
- *SimpleSokobanRandom:* * = up, & = down, 1 = left, M = right.

**Difficulty variations.** The environment can also be scaled to probe generalization under increased complexity. We train only on a $6 \times 6$ grid with a single box, and evaluate on:

- *LargerSokoban:* $10 \times 10$ grid with one box.
- *TwoBoxesSokoban:* $6 \times 6$ grid with two boxes.
- *ComplexSokoban:* $10 \times 10$ grid with two boxes (the hardest setting).

### 4.1.2 GENERAL POINTS

*General Points* (Chu et al., 2025) is an arithmetic reasoning game inspired by Point 24. The task requires using four given values exactly once to form an expression equal to a target value (default: 24).

**Instruction variations.** We vary the mapping of face cards (J, Q, K) to integers, testing whether the model generalizes beyond seen mappings. Training always uses J = Q = K = 10, while evaluation includes:

- *All 5:* J = Q = K = 5.
- *All 7:* J = Q = K = 7.

**Difficulty variations.** We further introduce two harder conditions:

- *Larger Number:* At least one input number is sampled from 14–19, which never appears in training.
- *Five Cards:* Input consists of five cards instead of four, increasing combinatorial complexity.

**Mixed variations.** We also evaluate combined instruction and difficulty shifts, where remapped face cards yield unseen numeric values ($> 10$). Each instance contains at least one face card:

- *All 12:* J = Q = K = 12.
- *Regular:* J = 11, Q = 12, K = 13 (as in Chu et al. (2025)).

## 4.2 DATA CURATION

### 4.2.1 ANSWER-ONLY DEMONSTRATIONS

For *Sokoban*, we use Breadth-First Search (BFS) to generate successful trajectories. For each state–action pair $(s, a)$ in these trajectories, we construct a problem prompt (Appendix B.1) that encodes the state $s$, the success criterion, and the action instructions. The expert action $a$ serves as the label. Expert demonstrations are collected only in the $6 \times 6$ single-box setting, yielding 3,981 state–action pairs.

For *General Points*, we adopt the dataset from Chu et al. (2025), randomly sampling 10,000 demonstrations from the full 800,000 for training.

### 4.2.2 CHAIN-OF-THOUGHT DEMONSTRATIONS

To construct chain-of-thought (CoT) demonstrations, we use the Qwen3-8B model (Yang et al., 2025), which is first RL-finetuned on each training task. Qwen3-8B is pretrained with reasoning-intensive data and STEM domains, and its post-training involves strong-to-weak distillation, making it a cost-effective alternative to human annotation. For every query $x \in \mathcal{D}$, we sample 16 candidate responses from the task-specific RL-finetuned Qwen3-8B. Rejection sampling is then applied to discard incorrect responses, and the remaining correct ones are retained as CoT demonstrations. This procedure yields a high-quality dataset of task-aligned reasoning traces.

## 5 REVISITING THE SFT MEMORIZATION MYTH

Prior work (Chu et al., 2025) compares SFT and RL across textual and visual domains, finding that while SFT achieves strong in-distribution (ID) performance, it often fails on out-of-distribution (OOD) settings with perturbed instructions or altered observations. By contrast, RL warm-started from an SFT checkpoint is more robust to such shifts, motivating the shorthand *"SFT memorizes, RL generalizes."* Jin et al. (2025) report similar results, focusing on how RL further shapes already SFT-tuned checkpoints. Complementarily, Xie et al. (2024) show that SFT trained with direct-answer supervision can *memorize* in ways that aid transfer across difficulty levels yet degrade sharply under instruction perturbations. Despite these observations, the precise mechanism by which SFT's "memorization" leads to OOD failures remains underexplored.

**Replication of prior findings.** We begin by reproducing the setup of Chu et al. (2025). As shown in Figure 1, SFT achieves solid in-distribution performance. However, under instruction variants, its accuracy typically rises briefly before collapsing to near zero. The sole exception is Qwen2.5-7B on *General Points*, which retains non-zero performance under instruction shifts.

**Instruction validity analysis.** Rather than concluding that SFT is fundamentally incapable of handling instruction variants, we disentangle *task competence* from *instruction adherence* using a validity metric. For *Sokoban*, a response is *valid* if the action tokens fall within the admissible set for the given variant (e.g., $\{1, 2, 3, 4\}$ in *SimpleSokobanNumerical*). For *General Points*, a response is *valid* if it applies the variant-specific face-card mapping correctly (e.g., J = Q = K = 5 in `All-5`) and yields a syntactically legal formula. As Figure 2 shows, SFT-tuned models initially exhibit high instruction validity, which then degrades rapidly during training. Only Qwen2.5-7B maintains moderate validity under *General Points* variants.

**Frozen-prompt hypothesis.** What explains this sharp drop in validity? We hypothesize that SFT trained on a single, frozen prompt template biases the policy toward a brittle surface mapping between tokens and actions. Because the training instructions never vary (e.g., fixed action names in *Sokoban*, or J = Q = K = 10 in *General Points*), the model overfits to training semantics and persists in using them even when faced with different instructions at test time.

**Testing with fake environments.** To verify this hypothesis, we construct *fake environments* that present *variant instructions* but continue to score responses using the *training semantics*. For example, in *FakeSokobanNumerical*, the prompt instructs the model to output numeric actions, yet the

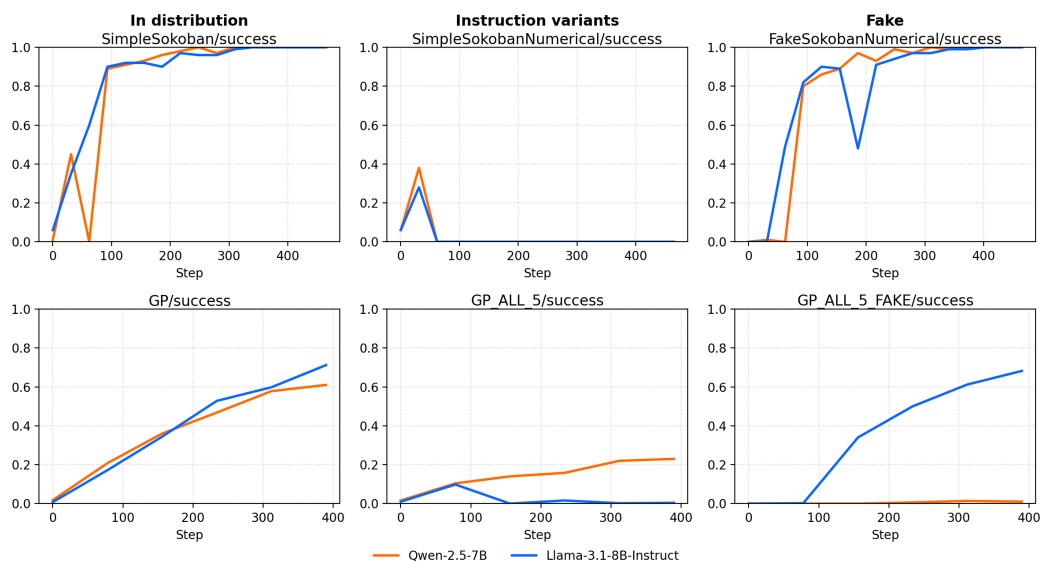

Figure 1: Success rates of SFT on *Sokoban* and *General Points*. Columns (left to right): in-distribution performance; instruction-variant performance; performance on the fake environment.

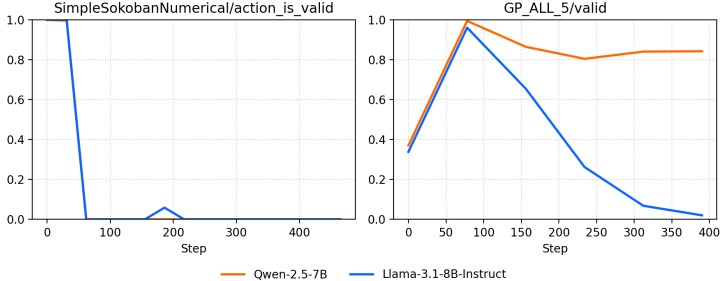

Figure 2: Instruction-following validity during SFT training on two instruction variants. **Left:** *SimpleSokobanNumerical*, where a response is valid if emitted action tokens lie in the admissible set $\{1, 2, 3, 4\}$. **Right:** *General Points (all_5)*, where validity requires correctly mapping J, Q, K to 5 and producing a legal equation.

environment only accepts the canonical tokens `up`, `down`, `left`, `right`. As shown in Figure 1 (right column), success in these fake environments closely tracks in-distribution performance—both on *Sokoban* and for Llama-3.1-8B-Instruct on *General Points*. This finding provides strong support for the frozen-prompt hypothesis.

## 5.1 DIVERSIFICATION SOLVES INSTRUCTION-VARIANTS

The evidence above suggests that SFT learns to map only the *varying* components of the input (e.g., the current *Sokoban* state or the set of cards in *General Points*) to solutions, while treating the *fixed* components of the prompt (action instructions in *Sokoban* and face-card mappings in *General Points*) as irrelevant. As a result, when these frozen components change at test time, the model continues to emit the training vocabulary and fails to generalize. To mitigate this issue, we introduce *prompt diversity* during training.

For *Sokoban*, we construct a vocabulary of random words. For each training example, four distinct words are uniformly sampled and used to replace the canonical action names (Up, Down, Left,

Right). The prompt explicitly specifies the mapping (e.g., "*w_1 means Up, w_2 means Down, w_3 means Left, w_4 means Right*"), forcing the model to rely on the declared mapping rather than memorized action tokens. For *General Points*, we diversify face-card interpretations by training under multiple mapping regimes, each indicated explicitly in the prompt. These include both uniform mappings (e.g., J = Q = K = 8 or 9) and staggered mappings (e.g., J = 7, Q = 8, K = 9), ensuring that the model sees a wide variety of valid interpretations during training.

As shown in Table 1, introducing prompt diversity markedly improves performance on unseen *Sokoban* instruction variants, while success on the *Fake* environments collapses—consistent with the frozen-prompt hypothesis. In-distribution accuracy remains high, with only minor changes relative to standard SFT. A similar pattern holds in *General Points* (Table 1b): diversity yields large gains on unseen variants, small improvements in in-distribution accuracy, and a sharp drop on the *Fake* split. Beyond diversifying prompts, we also apply light proximity control to keep the fine-tuned policy close to its base model and preserve instruction-following capability; detailed results are provided in Appendix **??**. While proximity control improves upon answer-only SFT, it is less effective than prompt diversification for instruction-variant robustness and can modestly degrade in-distribution performance and difficulty-variant generalization.

In summary, prompt diversity substantially improves instruction-variant generalization while preserving strong in-distribution performance. This intervention effectively resolves the instruction-following failures reported by Chu et al. (2025) and related studies. In effect, prompt diversity encourages the model to treat instructions as part of the input distribution to be interpreted, rather than fixed constants to be memorized.

# 6 CAN SFT GENERALIZE TO HARDER DOMAINS?

The previous section showed that prompt diversity resolves SFT's failures on instruction variants without sacrificing in-distribution accuracy. We now ask a stricter question: can SFT *scale* to harder regimes that demand reasoning and understanding of the tasks?

**Setup.** We evaluate generalization to harder regimes (see Section 4.1): *Sokoban* with larger grids and additional boxes(*LargerSokoban*, *TwoBoxesSokoban*, *ComplexSokoban*), and *General Points* with injected 14–19 values(*Large Number*) and five-card compositions(*Five Cards*).

**Answer-only SFT on difficulty variants.** As shown in Table 1 answer-only SFT produces policies that remain strong on the training distribution yet deteriorate markedly as tasks demand longer-horizon planning or greater compositional depth. Both backbones exhibit the same qualitative trend across all harder regimes in *Sokoban* (larger maps, more boxes) and *General Points* (larger magnitudes, five-card compositions), suggesting that the learned mapping is brittle to increases in search depth and combinatorial branching. This pattern implies reliance on shortcut solutions that fit the training manifold but do not instantiate transferable procedures. Practically, it underscores that improving instruction robustness via prompt diversity is not sufficient for scaling to difficulty: harder regimes require supervision that exposes intermediate structure (e.g., CoT) and likely benefit from proximity controls that avoid drifting into over-specialized solutions.

Recent evidence suggests that exposing intermediate reasoning improves transfer under hardness shifts. Xie et al. (2024) compare chain-of-thought (CoT) with answer-only supervision on difficulty variants in logical reasoning and find that CoT yields stronger generalization as problem difficulty increases. Motivated by these findings, we hypothesize that CoT will help SFT scale to our harder regimes.

**CoT SFT on difficulty variants.** As shown in Table 1, CoT supervision consistently improves generalization to harder regimes while preserving (and often enhancing) in-distribution performance. In *Sokoban*, the gains are most pronounced on settings that demand longer-horizon planning and multi-object coordination. In *General Points*, the strongest improvements occur on the most combinatorially challenging splits, where larger magnitudes and additional cards require deeper composition. Collectively, these trends indicate that CoT helps significantly with difficulty variants.

| Model | Method | ID | Alpha. | Num. | Rand. | Large | TwoBoxes | Complex | Fake |
|---|---|---|---|---|---|---|---|---|---|
| Qwen | Ans. | 0.98 | 0 | 0 | 0 | 0.64 | 0.35 | 0.19 | 0.93 |
| | Diver. + Ans. | 0.91 | 0.92 | 0.89 | 0.84 | 0.53 | 0.33 | 0.09 | 0 |
| | CoT | **1** | 0.73 | 0.8 | 0.22 | 0.53 | 0.57 | 0.28 | 0.03 |
| | Diver. + CoT | **1** | **0.97** | **0.98** | **1** | **0.74** | **0.58** | **0.4** | 0 |
| | RL (warm) | 0.9 | 0.76 | 0.69 | 0.55 | 0.34 | 0.36 | 0.1 | 0 |
| Llama | Ans. | 0.92 | 0 | 0 | 0 | 0.53 | 0.3 | 0.08 | 0.9 |
| | Diver. + Ans. | 0.91 | 0.89 | 0.91 | 0.85 | 0.47 | 0.18 | 0.12 | 0 |
| | CoT | **0.99** | 0.32 | 0.43 | 0.21 | 0.6 | **0.58** | 0.24 | 0.32 |
| | Diver. + CoT | **0.99** | **0.95** | **0.95** | **0.99** | **0.67** | **0.58** | **0.3** | 0 |
| | RL (warm) | 0.43 | 0.18 | 0.28 | 0.1 | 0.18 | 0.05 | 0.02 | 0 |

(a) *Sokoban*. Vertical rules separate ID — instruction variants (Alpha., Num., Rand..) — difficulty variants (*LargerSokoban*, *TwoBoxesSokoban*, *ComplexSokoban*) — Fake.

| Model | Method | ID | All-5 | All-7 | All-12 | Regular | Large | Five | Fake |
|---|---|---|---|---|---|---|---|---|---|
| Qwen | Ans. | 0.61 | 0.23 | 0.25 | 0.07 | 0.06 | 0.02 | 0.00 | 0.01 |
| | Diver. + Ans. | 0.78 | 0.71 | 0.70 | 0.09 | 0.07 | 0.03 | 0.01 | 0 |
| | CoT | 0.95 | 0.90 | 0.90 | 0.85 | 0.86 | 0.80 | 0.26 | 0 |
| | Diver. + CoT | **0.96** | **0.93** | **0.93** | **0.9** | **0.89** | **0.84** | **0.29** | 0 |
| | RL (warm) | 0.93 | 0.80 | 0.87 | 0.82 | 0.80 | 0.69 | 0.22 | 0.05 |
| Llama | Ans. | 0.71 | 0.00 | 0.00 | 0 | 0 | 0.02 | 0.01 | 0.68 |
| | Diver. + Ans. | 0.82 | 0.64 | 0.72 | 0.10 | 0.06 | 0.04 | 0 | 0 |
| | CoT | 0.96 | 0.91 | 0.93 | 0.83 | 0.86 | 0.80 | 0.18 | 0 |
| | Diver. + CoT | **0.97** | **0.93** | **0.93** | **0.9** | **0.89** | **0.82** | 0.23 | 0 |
| | RL (warm) | 0.92 | 0.87 | 0.83 | 0.72 | 0.76 | 0.65 | **0.37** | 0 |

(b) *General Points*. Vertical rules separate ID — instruction variants (*All-5*, *All-7*) — mixed (*All-12*, *Regular*) — difficulty (*LargeNumbers*, *FiveCards*) — Fake.

Table 1: Results for all methods on *Sokoban* and *General Points*. Entries are success rates (0–1). Training settings are detailed in Appendix C. We report the best checkpoints under the averaging these metrics except fake environments.

## 6.1 BEST OF BOTH WORLDS

While CoT supervision markedly improves performance on harder instances, Tables 1 show that CoT alone does *not* guarantee robustness to instruction variants. In *Sokoban*, CoT-tuned models still stumble when the action vocabulary is renamed and retain non-zero success on the *Fake* environments. Moreover, CoT lags behind *Diversity + answer-only* on instruction variants. These observations motivate **Diversity + CoT**: combining prompt diversity to solve instruction variants and CoT to solve difficulty-variants.

**Diversity + CoT achieves the best of both worlds.** As shown in Table 1, across both *Sokoban* and *General Points*, **Diversity + CoT** consistently outperforms alternative methods on instruction and difficulty variants—often by a wide margin (except FiveCards on General Points)—while maintaining strong in-distribution accuracy. Prompt diversity breaks surface anchoring (driving *Fake* success to near zero) by enforcing attention of the model to the instructions, and CoT supplies the intermediate-reasoning scaffold needed for difficulty generalization; either alone is insufficient, but together they match or surpass the strongest baselines. Compared with RL—which is exploration-dependent and optimization-unstable (especially from cold start, and still sensitive when warm-started)—learning can stall entirely when the base model cannot discover rewarding trajectories. In contrast, our approach attains comparable or better generalization without online exploration or reward-model design, preserving the stability and simplicity of supervised training and offering substantially better compute efficiency.

## 7 CONCLUSION

This paper revisits the generalization capacity of SFT and finds that, contrary to common belief, SFT can match or exceed RL) on our decision-making benchmarks when trained with the right data. We show that the widely cited "SFT memorization" phenomenon largely stems from frozen prompts; introducing prompt diversity breaks surface anchoring and yields instruction-invariant behavior. We further demonstrate CoT supervision transfer along difficulty axes. Combined, **Diversity + CoT** delivers a single, purely supervised recipe that is robust to instruction remappings and scales to harder regimes while maintaining strong in-distribution performance.

**Limitations and future work.** Our study only evaluates two tasks and two backbones; broader validation is warranted across modalities, longer-horizon interactive settings, and a wider range of base models. Another avenue is to strengthen SFT itself without data augmentation (e.g., objective shaping, proximal regularization(see Appendix D), improved selection/weighting), and to explore safe, compute-efficient hybrids that combine SFT's stability with RL's reward optimization—aiming for a practical best of both worlds in LLM training.

Overall, these results reframe the SFT–RL trade-off: with carefully designed data, *vanilla SFT* is a strong, practical default for post-training, capable of matching RL in generalization.

**Reproducibility Statement.** We provide anonymized source code and all datasets used in this work as supplementary materials, together with a detailed `README.md` that gives step-by-step instructions to reproduce our results. The package includes (i) dataset (including instruction, mixed, and difficulty variants) for both domains, (ii) training scripts and configuration files for all methods (answer-only SFT, Diversity, CoT, Diversity + CoT, and RL baselines) (iii) evaluation scripts.

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

## A  THE USE OF LARGE LANGUAGE MODELS (LLMS)

In this work, we utilized Large Language Models (LLMs) as a general-purpose assist tool for several specific tasks:

- **Paper Writing and Polishing**: We used LLMs to help refine and polish the writing of this paper, including improving clarity, grammar and typo correction.
- **Script Generation**: LLMs assisted in generating bash scripts to automate data processing, model training, and evaluation workflows.
- **Code Generation for Data Visualization**: We employed LLMs to generate code for creating data visualizations and analysis plots used in our experimental results.

The LLMs were used purely as assistive tools and did not contribute to the core research ideas, experimental design, or scientific insights presented in this work. All research contributions, experimental methodology, and conclusions remain entirely our own. The LLM assistance was limited to technical implementation support and writing enhancement, following standard academic practices for tool usage in research.

## B  PROMPT TEMPLATES

### B.1  SOKOBAN

The Sokoban environment challenges LLM agents to plan ahead to avoid dead-ends and achieve its goal of pushing boxes to designated locations. The prompt template follows initial implementation of (Wang et al., 2025). The following is the prompt template:

---

**Prompts**

```
<|im_start|>user
You are a Sokoban solver.
Sokoban Quick Guide
Goal:  Push all boxes (X) onto targets (O).
Symbols:
# Wall | _ Floor | O Target | X Box | P You | ✓ = Box on
Target | S = You on Target
Rules:
1.  Push boxes (can't pull).
2.  Avoid walls (#).
Answers:
<answer> Up </answer> | <answer> Down </answer> | <answer>
Left </answer> | <answer> Right </answer>
Rewards:
Move:  -0.1
Box on target:  +1.0
All boxes placed:  +10.0
[Current Observation]:{state information}
Decide the next action:
Always output:  <think> [Your thoughts] </think> <answer>
[your answer] </answer> with no extra text.  Strictly follow
this format.  <|im_end|>
<|im_start|>assistant
<think>
```

---

### B.2  GENERAL POINTS

The General Points environment challenges LLM agents to solve mathematical card games by creating equations that evaluate to a target number using four cards. The task requires both numerical reasoning and proper JSON formatting. The following is the prompt template:

```
Prompts

<|im_start|>user
[Task Description] You are an expert {target_number} points
card game player.  You will receive a set of {num_cards}
cards.  Note that {face_card_msg}, and each card must be used
once.  Your goal is to output a formula that evaluates to
{target_number} using numbers from the cards and operators
such as '+', '-', '*', '/', '(', ')', and '='.
[Input] Cards:  {cards}
[Output] { "cards":  [x, y, z, w], where {face_card_msg},
"number":  [a, b, c, d], where a, b, c, and d are the
numbers on the cards, "formula":  'an equation that equals
{target_number}', }
Always output:  <think> [Your thoughts] </think> <answer>
[your answer] </answer> with no extra text.  Strictly follow
this format.  <|im_end|>
<|im_start|>assistant
<think>
```

## C  EXPERIMENTAL SETUP

### C.1  TASK DESCRIPTION

We evaluate our approach on two challenging reasoning tasks: *Sokoban* and *General Points Equation*. Each task includes multiple subtask variations to test different aspects of generalization.

**Sokoban**  *Sokoban* is a classic puzzle game where the agent must push boxes onto target locations. The agent receives a grid-based observation and must output a sequence of actions to solve the puzzle.

Table 2: Sokoban Task Variations

| Variation Type | Subtask |
|---|---|
| Instruction | SimpleSokobanNumerical: 1=up, 2=down, 3=left, 4=right |
| | SimpleSokobanAlphabetical: A=up, B=down, C=left, D=right |
| | SimpleSokobanRandom: *=up, &=down, 1=left, M=right |
| Difficulty | LargerSokoban: $10 \times 10$ grid, 1 box |
| | TwoBoxesSokoban: $6 \times 6$ grid, 2 boxes |
| | ComplexSokoban: $10 \times 10$ grid, 2 boxes |
| Fake | FakeSokobanNumerical: Inconsistent prompt vs reward |

**General Points**  *General Points* is a mathematical reasoning task where the agent must use four cards to create an equation that evaluates to a target number (typically 24). Each card must be used exactly once with basic arithmetic operators.

**Task Configurations.** Both tasks support multiple environment configurations defined in the training system. For Sokoban, we use 8 different environment groups with varying grid sizes, box counts, and action mappings. For General Points, we support 5 different environment types with varying card counts, face card mappings, and difficulty levels.

**Evaluation Protocol.** Models are trained on the standard configurations and evaluated on all variation types to test instruction following, difficulty scaling. For *Sokoban*, success is considered as

Table 3: General Points Equation Task Variations

| Variation Type | Subtask |
|---|---|
| Instruction | All-5: J=Q=K=5 |
| | All-7: J=Q=K=7 |
| | All-12: J=Q=K=12 |
| Difficulty | Large Cards: At least one card from 14-19 |
| | Five Cards: 5 cards instead of 4 |
| Mixed | Face Cards as Regular: J=11, Q=12, K=13 |
| | Fake: Inconsistent prompt vs actual values |
| Training | Standard: J=Q=K=10, 4 cards, values 1-13 |

pushing all boxes to target location within 30 steps. For *General Points*, success is correctly recognized provided cards and output a correct formula that equals to target number.

## C.2 DATA

**SFT Training Data Generation**  We generate training data for both Sokoban and General Points Equation tasks using systematic approaches that ensure solution quality and task diversity.

For Sokoban, we generate training instances by creating solvable puzzle configurations and computing optimal solutions using search algorithms. The data generation process follows these steps:

1. **Puzzle Generation:** Create random Sokoban puzzles with varying grid sizes ($6 \times 6$ to $10 \times 10$) and box counts (1-2 boxes)
2. **Solution Verification:** Use depth-first search with a maximum depth of 30 steps to verify puzzle solvability
3. **Action Sequence Generation:** Extract the optimal action sequence that solves each puzzle
4. **Prompt Formatting:** Apply the Sokoban instruction template with grid observation and action mappings

For General Points Equation, we generate training data by creating solvable card combinations and computing valid mathematical solutions. The data generation process includes:

1. **Card Generation:** Randomly select required number of cards from 1-13 (A=1, J=11, Q=12, K=13)
2. **Face Card Mapping:** Apply predefined mappings (e.g., J=Q=K=10 for training)
3. **Solution Search:** Use brute force search to find valid equations that equal the target (24)
4. **Answer Formatting:** Generate structured JSON responses with cards, numbers, and formulas

The face card mapping system supports multiple configurations:

- **Training Mappings:** J=Q=K=10 (standard), mixed values (8,9,10), sequential patterns
- **Test Mappings:** All-5, All-7, All-12, Regular (J=11, Q=12, K=13)
- **Difficulty Variations:** 5-card games, large numbers (14-19), fake prompts

Both data generation processes ensure high-quality training data through:

- **Solution Verification:** All generated instances have verified solutions
- **Diverse Configurations:** Multiple environment variants and face card mappings

- **Balanced Difficulty:** Mix of easy and challenging instances
- **Consistent Formatting:** Standardized prompt templates and answer formats

The generated datasets are saved in Parquet format and organized into training and validation splits, with separate instances for supervised fine-tuning (SFT) and reinforcement learning (RL) training phases.

**Chain-of-Thoughts**   We adopt a reinforcement-learning trained **Qwen3-8B** models to generate Chain-of-Thoughts using prompts from above data generation process.

**Reinforcement Learning Training Data Generation**   For reinforcement learning training, we use the same underlying task configurations as SFT but with a different data format and generation approach.

RL training data differs from SFT in several key ways:

- **No Ground Truth Answers:** Unlike SFT, RL data contains only the input prompts without pre-computed solutions
- **Environment Integration:** Data is directly consumed by the RL environment for real-time interaction and reward computation

## C.3   TRAINING PIPELINE

**SFT**   We use a FSDP (Fully Sharded Data Parallel) training framework built on top of ver (2024) for efficient large-scale model training.

The SFT training configuration uses the following key hyperparameters and settings:

Table 4: SFT Training Hyperparameters

| Parameter | Value |
|---|---|
| Learning Rate | $1 \times 10^{-5}$ |
| Batch Size | 128 |
| Micro Batch Size per GPU | 1 |
| Max Response Length | 7000 |
| Training Epochs | 15 |
| Optimizer | AdamW |
| Weight Decay | 0.01 |
| Gradient Clipping | 1.0 |

**Reinforcement Learning**   We choose to use **GRPO** Shao et al. (2024) for our RL training algorithm with the following recipe:

The base models struggle to achieve positive rewards without any supervised fine-tuning (SFT). Therefore, we adopt a two-stage training approach where we first perform SFT on chain-of-thought data with 10 steps, then use these SFT checkpoints as warm-up models for reinforcement learning training. This approach ensures that the models have learned the basic task structure and reasoning patterns before being exposed to the reward-based learning environment.

## C.4   ENVIRONMENT SETUP AND REWARD DESIGN

**Environment Configuration**   We implement modular environment systems for both tasks with configurable parameters and multiple variants. The Sokoban environment supports configurable grid dimensions with different number of boxes, featuring four directional actions (Up, Down, Left, Right) across 8 different configurations including SimpleSokoban, LargerSokoban, ComplexSokoban, and various instruction formats (Numerical, Alphabetical, Random).

In contrast, the General Points environment focuses on mathematical reasoning with variable card counts (4-6 cards), configurable face card mappings, and a default target value of 24. Unlike

Table 5: RL Training Hyperparameters

| Parameter | Value |
|---|---|
| Learning Rate | $1 \times 10^{-6}$ |
| Batch Size | 256 |
| Mini Batch Size | 256 |
| Micro Batch Size per GPU | 64 |
| Max Prompt Length | 1000 |
| Max Response Length | 7000 |
| KL Coefficient | 0.0 |
| Entropy Coefficient | 0.0 |
| Gradient Clipping | 1.0 |
| Rollout Workers | 8 |
| GPU Memory Utilization | 0.7 |
| Tensor Parallel Size | 1 |
| Total Training Epochs | 100 |

Sokoban's multi-turn design, GP-L tasks are structured as single-turn problems requiring complete solutions in one response.

**Reward Function Design** The reward systems are tailored to each task's specific requirements. For *Sokoban*, we implement a binary reward structure where the model receives a score of 1.0 for correct actions (matching the ground truth expert action) and 0.0 for incorrect actions, with an additional format score of 0.1 for properly formatted but incorrect responses. The system features dynamic action mapping based on instruction variants and provides immediate feedback to guide learning.

The *General Points* reward system emphasizes mathematical accuracy with a structured scoring approach: +5 points for correct solutions that equal the target, +1 point for partial solutions that use valid numbers but don't reach the target, and progressive penalties: -2 for using invalid numbers or incorrect number counts, and -3 for illegal formula syntax. The system includes comprehensive validation to ensure proper JSON structure and mathematical correctness.

## D  ADDITIONAL EXPERIMENT RESULTS

In this section, we provide additional experimental results which are not covered in the main body. We think those results are very insightful to help us understand how SFT generalizes with regularization techniques, specifically L2 anchor regularization and KL divergence regularization. These regularization methods help prevent the model from deviating too far from the base model during fine-tuning, which is crucial for maintaining generalization capabilities while learning task-specific patterns.

The design principle of this experiment is to understand how regularization impacts SFT generalization. We systematically evaluate the effect of different regularization strengths on model performance across in-distribution and out-of-distribution test cases. Our experimental setup includes the following configurations:

- **SFT**: Standard supervised fine-tuning using vanilla cross-entropy loss $\mathcal{L}_{SFT}$.
- **SFT+$\alpha$KL**: Augmented with KL divergence regularization, where the total loss becomes:
$$\mathcal{L} = \mathcal{L}_{SFT} + \alpha \cdot D_{KL}(\pi_\theta \| \pi_{ref}) \tag{1}$$
  where $D_{KL}(\pi_\theta \| \pi_{ref}) = \mathbb{E}_{(x,y)\sim\mathcal{D}} \left[ \sum_t \pi_\theta(y_t|x,y_{:t-1}) \log \frac{\pi_\theta(y_t|x,y_{:t-1})}{\pi_{ref}(y_t|x,y_{:t-1})} \right]$ measures the KL divergence between the trainable model $\pi_\theta$ and the reference base model $\pi_{ref}$.
- **SFT+$\alpha$L$_2$**: Augmented with L2 anchor regularization, where the total loss becomes:
$$\mathcal{L} = \mathcal{L}_{SFT} + \alpha \cdot \|\theta - \theta_{ref}\|_2^2 \tag{2}$$
  where $\theta_{ref}$ represents the parameters of the base model and $\|\theta - \theta_{ref}\|_2^2 = \sum_i (\theta_i - \theta_{ref,i})^2$ penalizes large deviations from the reference parameters.

| | | Sokoban | | | |
|---|---|---|---|---|---|
| Base Model | Method | In-distribution | Larger | TwoBoxes | Complex |
| Llama | SFT | 0.92 | 0.38 | 0.17 | 0.05 |
| | SFT+0.05KL | 0.97 | 0.44 | 0.26 | 0.08 |
| | SFT+0.1KL | 0.91 | 0.54 | 0.33 | 0.09 |
| | SFT+0.5KL | 0.78 | 0.30 | 0.16 | 0.06 |
| | SFT+0.05L2 | 0.93 | 0.50 | 0.24 | 0.07 |
| | SFT+0.1L2 | 0.81 | 0.38 | 0.16 | 0.04 |
| | SFT+0.5L2 | 0.77 | 0.30 | 0.29 | 0.03 |
| Qwen | SFT | 0.99 | 0.56 | 0.41 | 0.18 |
| | SFT+0.05KL | 0.97 | 0.44 | 0.26 | 0.08 |
| | SFT+0.1KL | 0.91 | 0.54 | 0.33 | 0.09 |
| | SFT+0.5KL | 0.95 | 0.55 | 0.37 | 0.15 |
| | SFT+0.05L2 | 0.99 | 0.53 | 0.31 | 0.07 |
| | SFT+0.1L2 | 0.98 | 0.50 | 0.36 | 0.09 |
| | SFT+0.5L2 | 0.84 | 0.41 | 0.23 | 0.03 |

Table 6: Success rates on Sokoban task with different regularization methods. In-distribution = 6×6–1B (training config), Larger = 10×10–1B, TwoBoxes = 6×6–2B, and Complex = 10×10–2B. Values are success rates (0-1).

| | | General Points | | |
|---|---|---|---|---|
| Base Model | Method | In-distribution | Larger Numbers | Five Cards |
| Llama | SFT | 0.712 | 0.016 | 0.006 |
| | SFT+0.05KL | 0.658 | 0.018 | 0.008 |
| | SFT+0.1KL | 0.648 | 0.026 | 0.004 |
| | SFT+0.5KL | 0.104 | 0.004 | 0.000 |
| | SFT+0.01L2 | 0.560 | 0.036 | 0.036 |
| | SFT+0.05L2 | 0.336 | 0.032 | 0.054 |
| | SFT+0.1L2 | 0.230 | 0.036 | 0.034 |
| | SFT+0.5L2 | 0.152 | 0.032 | 0.046 |
| Qwen | SFT | 0.610 | 0.018 | 0.002 |
| | SFT+0.05KL | 0.610 | 0.010 | 0.004 |
| | SFT+0.1KL | 0.572 | 0.020 | 0.006 |
| | SFT+0.5KL | 0.286 | 0.024 | 0.010 |
| | SFT+0.01L2 | 0.420 | 0.038 | 0.012 |
| | SFT+0.05L2 | 0.236 | 0.052 | 0.004 |
| | SFT+0.1L2 | 0.166 | 0.050 | 0.008 |
| | SFT+0.5L2 | 0.098 | 0.034 | 0.014 |

Table 7: Success rates on General Points task with different regularization methods. In-distribution uses standard 4-card setup, Larger Numbers includes values 14-19, and Five Cards uses 5 cards instead of 4. Values are success rates (0-1).

We evaluate each configuration with regularization coefficients $\alpha \in \{0.05, 0.1, 0.5\}$ to understand the trade-off between task-specific learning and maintaining proximity to the base model. The KL regularization encourages the model to maintain similar output distributions to the reference model, while L2 regularization constrains the parameter space directly. Both approaches aim to preserve the general knowledge encoded in the base model while allowing task-specific adaptation.

**Training Recipe**  We resue the same training recipe to Table 4 except that training epochs is 5.