# OpenReview forum: "Debunk the Myth of SFT Generalization"
_ICLR.cc/2026/Conference — ICLR 2026 Conference Withdrawn Submission_

### Official Review · Reviewer_Nsz4 · 2025-10-15

**Soundness:** 2
**Presentation:** 3
**Contribution:** 2
**Rating:** 4
**Confidence:** 4

**Summary:**

This paper challenges this myth through systematic experiments on two decision-making benchmarks—Sokoban (grid puzzle) and General Points (arithmetic reasoning)—and reveals that SFT’s perceived generalization failure stems not from its maximum-likelihood objective, but from frozen-prompt artifacts.

**Strengths:**

- The paper’s originality lies not in inventing entirely new algorithms, but in reframing a long-standing problem, removing limitations of prior work, and creatively combining existing ideas for novel purposes.

- The paper does not invent prompt diversity or CoT in isolation, but combines them to solve complementary gaps—a creative synthesis that outperforms individual methods and RL

**Weaknesses:**

- The paper’s validation is restricted to closed-set decision tasks (Sokoban: fixed grid goals; General Points: fixed target value of 24) with objective "correct/incorrect" labels. This design fails to address the most common real-world scenarios where SFT is applied—open-domain generation (e.g., story writing, summarization) or subjective tasks

- The paper describes prompt diversity as "randomly sampling words for Sokoban actions" or "mixing face-card mappings" but does not answer:
  - For Sokoban: How many unique action-vocabulary mappings are needed to ensure generalization? (e.g., 5 vs. 20 variants—too few may not break frozen-prompt habits; too many inflates data costs.)
  - For General Points: What percentage of training data should use non-default face-card mappings? (e.g., 30% vs. 70%—too little may leave the model biased toward training semantics.)

- The paper empirically shows that "Diversity + CoT" improves generalization but provides no theoretical or mechanistic insight into how these interventions change the model’s learning behavior.

- It seems odd to sample CoT data from a model refined by RL, then use that data to SFT another model, and finally claim that the SFT model surpasses the original RL-trained model.

**Questions:**

see weakness

---

### Official Review · Reviewer_9LRw · 2025-10-25

**Soundness:** 2
**Presentation:** 3
**Contribution:** 1
**Rating:** 2
**Confidence:** 4

**Summary:**

This paper challenges the belief that supervised fine-tuning (SFT) cannot generalize, showing that failures largely stem from frozen prompt artifacts rather than the SFT objective itself.
By introducing prompt diversity and chain-of-thought (CoT) supervision, the authors demonstrate that plain SFT can match or exceed RL-based fine-tuning on decision-making benchmarks.
The work reframes SFT vs. RL as a data-centric trade-off, arguing that carefully curated demonstrations, not algorithmic complexity, drive generalization.

**Strengths:**

- Provides strong empirical evidence that SFT can generalize as well as RL when trained on well-designed data.
- Introduces simple, effective fixes, prompt diversity and chain-of-thought supervision instead of complex algorithms.
- Easy-to-understand writing and well-structured presentation of the paper.

**Weaknesses:**

This is a very interesting work; However, the fact that **the authors used a synthetic dataset** to argue that "SFT can also match the performance of RL" in LM post-training critically weakens the message:

1) I believe that the community’s post-training debate is fundamentally about **sample efficiency under limited coverage**, i.e., whether RL can generalize better when a **fixed dataset cannot span the long-tail, open-domain distribution** (which occurs in most cases), rather than about the intrinsic superiority of one **learning objective** over another. Prior work in RL vs. behavior cloning is already well studied in RL domain and shows that, with sufficiently broad and diverse data, imitation can approach RL performance, while being more vulnerable to covariate shift and spurious correlations. Against that backdrop, it’s unsurprising that with ample prompt diversity and CoT supervision, SFT can match RL on closed, synthetic tasks.

2)  The question of “Which objective generalizes better?” in LM post-training is entangled with **how models are initialized** and **what data/priors they inherit**. Warm starts from strong demonstrations and large-scale **pretraining** inject powerful priors that can dominate outcomes. In this manner, I believe the study should be better framedand evaluated in a more **data-centric** way that explicitly controls for pretraining, teacher signals and warm-start asymmetries.

3) Because the paper evaluates only **decision-making** benchmarks (Sokoban, arithmetic card games) and explicitly does not cover open-ended/creative language generation, its conclusions offer limited practical guidance for the open-domain settings that motivate the sample-efficiency debate.

4) If the focus is on more SFT-vs-RL comparison on synthetic data, it could be better to run it in a fully controlled setting, i.e. by training a vanilla transformer from scratch. However, I'm afraid then making implications to "LM post-training" might be a little challenging.

**Questions:**

- Why did you pair synthetic decision-making (and were there any motivations for specifically choosing these tasks) benchmarks with pre-trained models?

- Were there any attempts to explore this in open-domain generation and what were some difficulties you found during the course of action?

- Please see weaknesses section. If the authors could resolve these concerns, I am willing to reconsider my score.

---

> ### Author Response · Authors · 2025-11-14
>
> Dear reviewer 9LRw,
>
> Thank you again for your thoughtful and constructive feedback.
> We are writing to better understand weakness #3, where you suggest that open-domain tasks may provide a more suitable testbed for studying the generalization behavior of SFT vs. RL. This point stood out to us because it differs from how recent literature has framed the comparison between these two paradigms.
>
> Context for our question.
>
> Our paper is motivated by and serves as a reflection on a recent line of work [1–7] arguing that SFT underperforms RL in out-of-distribution generalization. Among them, [1,2,3] focus on decision-making environments (e.g., GeneralPoints, visual navigation, KnightsKnaves), while [4.5.6.7] focus on reasoning tasks (e.g., math or symbolic deduction).
>
> While these domains differ, a key commonality is that they all provide verifiable rewards or labels, allowing (1) Vanilla RL (not RLHF) to be applied directly, and (2) Unbiased, objective evaluation of the resulting policies. Because this property makes the SFT–RL comparison methodologically clean, our intuition has been that structured decision-making tasks with verifiable reward are a natural experimental setting for studying generalization differences.
>
> For this reason, we are very interested in understanding your perspective on open-domain tasks. Could you help us clarify:
> What aspects of open-domain settings make them, in your view, more diagnostic or more revealing for distinguishing the generalization behaviors of SFT and RL?
>
>
> Are there specific prior works that support or motivate this expectation?
>
>
> How do you envision handling the lack of a verifiable reward signal in open-ended tasks—something that past SFT vs RL comparisons [1–7] generally avoid due to evaluation and bias concerns?
> We ask these questions not to challenge your point, but because we genuinely want to understand the underlying reasoning. Your insight may help us revise our framing in a more coherent and broadly useful way.
> Thank you again for your time and thoughtful review.
>
> [1] Chu, Tianzhe, et al. "Sft memorizes, rl generalizes: A comparative study of foundation model post-training." arXiv preprint arXiv:2501.17161 (2025).
>
> [2] Jin, Hangzhan, et al. "Rl fine-tuning heals ood forgetting in sft." arXiv preprint arXiv:2509.12235 (2025).
>
> [3] Xie, Chulin, et al. "On memorization of large language models in logical reasoning." arXiv preprint arXiv:2410.23123 (2024).
>
> [4] Wu, Yongliang, et al. "On the generalization of sft: A reinforcement learning perspective with reward rectification." arXiv preprint arXiv:2508.05629 (2025).
>
> [5] Zhu, Wenhong, et al. "Proximal supervised fine-tuning." arXiv preprint arXiv:2508.17784 (2025).
>
> [6] Huan, Maggie, et al. "Does Math Reasoning Improve General LLM Capabilities? Understanding Transferability of LLM Reasoning." arXiv preprint arXiv:2507.00432 (2025).
>
> [7] Kang, Katie, et al. "What Do Learning Dynamics Reveal About Generalization in LLM Reasoning?." arXiv preprint arXiv:2411.07681 (2024).

---

> ### Comment · Reviewer_9LRw · 2025-11-15
>
> Dear authors,
>
> Thanks for the thoughtful note. To clarify my earlier point: by “open-domain” (not *open-ended*) I don’t mean unconstrained, free-form generation without ground truth. I mean natural-language, more diverse and realistic tasks where inputs/outputs are free text but success is still programmatically checkable: i.e. code generation/repair with unit tests, text-to-SQL with exact or execution match, complex math word problems with exact answers, or web agents with clear task completion validators. These stay within the verifiable-reward spirit you emphasize, but they stress the kinds of linguistic variability (paraphrase, formatting, tool/API calls, compositionality) where we often see practical gains or failures in post-training.
>
> I also understand that your study follows the RL-vs-SFT line of work in RL with Verifiable Rewards (RLVR). However, RLVR doesn’t naturally extend to genuinely open-ended language use, where you’d need a learned reward model or LLM-as-a-judge, and those bring evaluation bias and robustness issues despite efforts like RLAIF. So if the paper is framed broadly as “RL vs SFT” in language modeling, it can read as over-generalizing from RLVR. If that’s not your intent, I’d encourage making it explicit that your claims are about RLVR vs SFT, not RL in general.
>
> One more note on datasets and initialization: many demonstrations of RL gains assume tasks that live reasonably close to the model’s pretraining distribution. Pairing a synthetic environment with a pretrained LM can blur interpretation, strong priors and warm-starts may dominate, and results can hinge on prompt artifacts rather than the learning objective. This is why I suggested language-first RLVR tasks with automatic checks: they at least resemble practical use and directly test whether your data-centric SFT recipe holds up under realistic linguistic variation, while in this case your message should be reframed in a little different way rom now, but I feel like this at least delivers a clear, practical takeaway that challenges recent claims about SFT’s poor generalization.

---

> > ### Author Response · Authors · 2025-11-15
> >
> > Thank you for the quick response! The clarification is very helpful. We will improve our paper along the lines you and other reviewers have raised. Thanks again for your time and effort in reviewing our manuscript.

---

### Official Review · Reviewer_zxhg · 2025-11-01

**Soundness:** 3
**Presentation:** 2
**Contribution:** 2
**Rating:** 4
**Confidence:** 4

**Summary:**

This paper reexamines the belief that supervised fine-tuning (SFT) memorizes and fails to generalize compared to reinforcement learning (RL). Through Sokoban and General Points benchmarks, it shows that SFT’s reputed brittleness largely stems from “frozen-prompt” artifacts: training on fixed instruction templates causes models to ignore instruction remappings. Using “Fake” environments, the authors causally demonstrate this prompt anchoring. They propose a simple, data-centric recipe: prompt diversity (varying instruction mappings during training) to handle instruction variants, and chain-of-thought (CoT) supervision to transfer to harder regimes (larger grids/boxes, larger numbers, five-card compositions). Combined, Diversity + CoT delivers strong in-distribution and out-of-distribution performance, often matching or surpassing warm-start RL baselines.

**Strengths:**

- This paper implements multiple OOD variants of Sokoban and GP-L, which provides a more comprehensive assessment of models' generalization capability.
- Simple and neat finding. This paper narrows down the empirical benefit of SFT data construction to CoT and diversified prompts and provides analysis on it.

**Weaknesses:**

- Several recent papers [1, 2, 3] have shown that SFT on small, high-quality datasets—particularly via distillation of long chain-of-thought from stronger “thinking” models—can substantially improve math/reasoning performance. Given this context, the incremental empirical novelty of the present work may be limited. You can easily reach high-scores (both ID and OOD) on easy synthetic tasks like GP-L using similar distillation procedures as it's much simpler than real-world math.

- Much of the paper’s gains appear to come from data-centric interventions (prompt diversity and CoT), which could be interpreted as a form of data augmentation and prompt design. The title and narrative might better reflect this emphasis—e.g., by foregrounding the data-design perspective on SFT generalization—so readers do not infer that the improvements stem from algorithmic advances in SFT itself.

- Experiments are not well controlled.
  - In SFT: diversified prompt, CoT are introduced
  - In RL: only one vanilla training setting
  - It's unfair to compare SFT and RL in this setting as equivalent data variations are not introduced in RL training.
    - You may consider ablating warming up RL checkpoints using these data variations.

- Sokoban and GP-L may not be difficult enough today. As reported in Table 1, Diver. + CoT reaches >0.9 accuracy. More difficult evaluation settings should be considered to strengthen your claim.

- Minor issue: missing reference in line 337  Appendix ??

[1] Ye, Yixin, Zhen Huang, Yang Xiao, Ethan Chern, Shijie Xia, and Pengfei Liu. "Limo: Less is more for reasoning." arXiv preprint arXiv:2502.03387 (2025).

[2] Muennighoff, Niklas, Zitong Yang, Weijia Shi, Xiang Lisa Li, Li Fei-Fei, Hannaneh Hajishirzi, Luke Zettlemoyer, Percy Liang, Emmanuel Candès, and Tatsunori Hashimoto. "s1: Simple test-time scaling." arXiv preprint arXiv:2501.19393 (2025).

[3] Wang, Zengzhi, Fan Zhou, Xuefeng Li, and Pengfei Liu. "Octothinker: Mid-training incentivizes reinforcement learning scaling." arXiv preprint arXiv:2506.20512 (2025).

**Questions:**

- Is prompt diversity introduced in RL training? I did not see any paragraphs mentioning this. If not, it would be a bit unfair to compare SFT and RL because they are trained on different data.
- What's the statistics of the constructed diversified prompts? Take GP-L as an example, what's the proportion of {J=Q=K=10, J=Q=K=7, J=7 Q=8 K=9}?
- Your prompt diversification strategy introduces new data samples into the training data. What if you diversify non-task-related prompts, i.e. synthetically collecting prompt templates while maintaining the original task related information?
- CoT + diverse prompts + SFT yields the highest performance. What if the RL training starts from the early training stage of this SFT process (hence the RL model can also do CoT)?

---

### Official Review · Reviewer_QtPF · 2025-11-01

**Soundness:** 1
**Presentation:** 3
**Contribution:** 1
**Rating:** 2
**Confidence:** 4

**Summary:**

This paper discusses the belief that Supervised Fine-Tuning (SFT) lacks generalization on decision-making tasks, compared to Reinforcement Learning (RL). The experiment results first show that the most basic SFT (without any instruction perturbation) will suffer huge performance drop with instruction variations, which align with prior work observation. The further experiments demonstrate by applyng diverse instructions and chain-of-thoughts during SFT can achieve performance surpassing RL baseline.

**Strengths:**

1. The SFT generalization problem is an interesting topic to discuss. Since SFT and RL are both prominent methods for post-training, it is interesting to discuss the pros and cons of each.
2. The narrative of the paper is straightforward and easy to follow. The writing is clear.

**Weaknesses:**

Currently, my main concern is about the lack of experiment details and the naive experiment design, which makes it difficult to believe that the findings of this paper is solid and can generalize to other tasks.
1. The experiment setting is largely missing from the main paper. Even when I read the Appendix, it's still unclear to me what's the base model the author is using for SFT and RL in Table 1. Can you clarify what model are you using?
2. Following question 1, from fig 1, I'm assuming the base model, such as qwen you're using for SFT and RL, is qwen2.5-7B. If so, the SFT + COT results is unfair to compare with RL, since COT data are generated by a stronger model Qwen3-8B. This make the entire argument of SFT can outperform RL with simply COT + Diverse Input very unpersuasive. I'll recommend the authors to provide the Qwen-3-8B results of these two benchmark, for a fair comparison. The SFT + COT + Diverse Instruction needs to further surpass Qwen-3-8B performance, otherwise it's simply learning from a better model and can easily outperform the RL baseline.
3. Following 2., can the author provide the hyper-param search results for RL and SFT baselines, to make sure that these baselines are also well trained.
4. The observations of SFT requires COT and diverse instruction to achieve better generalizibility is already seen in most instruction tuning papers. For example, Flan-V2 [1] already applies it a few years ago and is a standard for instruction-tuning SFT now.
5. The conclusion of SFT generalize better than RL on decision making tasks when trained with COT + Diverse instruction might not generalize to other general reasoning tasks, making the constribution very limited. Will recommend the author to test on at least one reasoning tasks. (For example, math dataset. Recommend the DAPO [2] setting or LIMR [3] if you lack gpu resource.


[1] The Flan Collection: Designing Data and Methods for Effective Instruction Tuning

[2] DAPO: An Open-Source LLM Reinforcement Learning System at Scale

[3] LIMR: Less is More for RL Scaling

**Questions:**

1. Line 41 and 117 mention the problem of collapse output diversity of SFT. However, this is actually the mode collapse problem of RL, too.   Recent work actually shows that RL reduce more of the output diversity compared to SFT [1, 2, 3]. Also this paper does not discuss the output diversity of RL and SFT with the experiment results, will recommend the author ro remove these statements.

2. Line 337: Appendix reference is missing.


[1] Beyond Reverse KL: Generalizing Direct Preference Optimization with Diverse Divergence Constraints

[2] Diverse Preference Optimization

[3] Diverse Preference Learning for Capabilities and Alignment

---

### Note · Authors · 2025-11-30

**Comment:**

We sincerely thank all reviewers for the time and care put into evaluating our manuscript. We greatly appreciate the thoughtful and constructive feedback, which has provided us with valuable insights for improving the rigor and clarity of our work.
We are grateful for the reviewers’ contributions, and we will carry these recommendations forward as we refine the project.
Thank you again for your effort and expertise.

**Withdrawal Confirmation:**

I have read and agree with the venue's withdrawal policy on behalf of myself and my co-authors.